# NAFLD and NAFLD Related HCC: Emerging Treatments and Clinical Trials

**DOI:** 10.3390/ijms26010306

**Published:** 2025-01-01

**Authors:** Tripti Khare, Karina Liu, Lindiwe Oslee Chilambe, Sharad Khare

**Affiliations:** 1Division of Gastroenterology and Hepatology, Department of Medicine, University of Missouri, Columbia, MO 65212, USA; kharet@health.missouri.edu; 2Harry S Truman Memorial Veterans’ Hospital, Columbia, MO 65201, USA; 3Department of Biochemistry and Molecular Biology, Bloomberg School of Public Health, Johns Hopkins University, Baltimore, MD 21205, USA; kliu100@jh.edu; 4School of Medicine, Copperbelt University, Kitwe P.O. Box 21692, Zambia; lindychilambe36@gmail.com

**Keywords:** nonalcoholic fatty liver disease, nonalcoholic steatohepatitis, hepatocellular carcinoma, cirrhotic and non-cirrhotic HCC, treatments, clinical trials

## Abstract

Nonalcoholic fatty liver disease (NAFLD), recently renamed metabolic-associated fatty liver disease (MAFLD), is the most prevalent liver disease worldwide. It is associated with an increased risk of developing hepatocellular carcinoma (HCC) in the background of cirrhosis or without cirrhosis. The prevalence of NAFLD-related HCC is increasing all over the globe, and HCC surveillance in NAFLD cases is not that common. In the present review, we attempt to summarize promising treatments and clinical trials focused on NAFLD, nonalcoholic steatohepatitis (NASH), and HCC in the past five to seven years. We categorized the trials based on the type of intervention. Most of the trials are still running, with only a few completed and with conclusive results. In clinical trial NCT03942822, 25 mg/day of milled chia seeds improved NAFLD condition. Completed trial NCT03524365 concluded that Rouxen-Y gastric bypass (RYGB) or sleeve gastrectomy (SG) results in histological resolution of NASH without worsening of fibrosis, while NCT04677101 validated sensitivity/accuracy of blood biomarkers in predicting NASH and fibrosis stage. Moreover, trials with empagliflozin (NCT05694923), curcuvail (NCT06256926), and obeticholic acid (NCT03439254) were completed but did not provide conclusive results. However, trial NCT03900429 reported effective improvement in fibrosis by at least one stage, without worsening of NAFLD activity score (NAS), as well as improvement in lipid profile of the NASH patients by 80 or 100 mg MGL-3196 (resmetirom). Funded by Madrigal Pharmaceuticals, Rezdiffra (resmetirom), used in the clinical trial NCT03900429, is the first FDA-approved drug for the treatment of NAFLD/NASH.

## 1. Introduction

Nonalcoholic fatty liver disease (NAFLD) and particularly its progressive disease stage, nonalcoholic steatohepatitis (NASH), is the most common liver disease worldwide that affects almost 25% of the global population [1]. This number is anticipated to rise to 56% in Europe, USA, and China in the next ten years [2]. NAFLD is an array of liver diseases that occur particularly in the absence of excess alcohol use or other known causes. NAFLD is associated with fatty liver (with fat contributing to more than 5% of liver weight) and metabolic dysfunction. A group of experts recently changed the terminology NAFLD to metabolic associated fatty liver disease (MAFLD) to more accurately account for pathogenesis and patient stratification for the management of NAFLD [3]. However, in the present article we have still used term NAFLD instead of MAFLD to accommodate all the prior discoveries and clinical trials. Hepatocellular carcinoma (HCC) is the fatal complication of NAFLD-NASH [4]. HCC, accounting for 70% to 80% of all primary liver cancers, is currently among the most rapidly rising causes of cancer-related deaths in the United States [5,6]. It is the third leading cause of cancer-related deaths globally, potentially displaying an increase in mortality rates in multiple areas of the western world [7,8]. HCC usually occurs within an established background of chronic liver disease and cirrhosis [6], although it can also occur in the absence of cirrhosis [9]. The development of HCC from non-cirrhotic livers comprises up to 54% of all HCC cases and growing literature attributes this occurrence to NAFLD [10,11,12].

The outcome for patients with HCC remains dismal regardless of current developments and breakthroughs [13] unless patients receive an early diagnosis or they are able to undergo curative treatment [14]. Liver cancer has some distinct qualities as a neoplasm, which grows on the background of cirrhosis, and is substantially more resistant to conventional chemotherapy. The HCC surveillance system follows a distinct non-invasive diagnostic criterion [15]. Surveillance is usually recommended for patients identified as high risk to mitigate the disease burden and increase the proportion of patients detected at early stages [14]. In China during the 1900s, a randomized clinical trial for 18,816 patients with hepatitis B virus (HBV), comparing patients who were surveilled using ultrasound and α-fetoprotein to those who were not, showed a significant increase in the early detection of HCC and receipt of curative treatment [7,16]. This resulted in a 37% reduction in HCC-related mortality [7]; however, it is unclear whether these data are generalizable to NAFLD-related cirrhosis. Professional societies have different recommendation guidelines for HCC surveillance. American Association For The Study of Liver Diseases (AASLD) and European Association For The Study Of The Liver (EASL) recommend that semi-annual abdominal ultrasound be conducted with or without α-fetoprotein as the primary strategy for HCC surveillance [17,18]. For advanced HCC, which is usually nonresectable, the current treatment guidelines recommend the use of sorafenib and lenvatinib as first-line treatment and regorafenib, cabozantinib, and ramucirumab as second-line treatment [15]. In patients with HCC and microvascular invasion (MVI), long-term prognosis is improved by postoperative adjuvant therapy with lenvatinib [19]. Furthermore, the use of oral lenvatinib can lower the risk of tumor relapse and promote long-term survival of patients with HCC and MVI [19].

Plenty of reviews on NAFLD and HCC were published in the past. However, not many focus on the clinical trials being performed in relation to NAFLD/NASH and HCC. In this review, we chose to focus primarily on clinical trials over the past five to seven years to summarize the experiments being performed to treat or better diagnose, as well as surveil and screen, NAFLD, NASH, and/or HCC. We used PubMed to find references mostly published in the last five to seven years, with the keywords “NAFLD”, “NASH”, “HCC”, and “clinical trials”. Clinical trials were then searched on the website clinicaltrials.gov using the keywords “HCC—hepatocellular carcinoma” and “NAFLD—nonalcoholic fatty liver disease”. Google Scholar was used to research targets mentioned in the clinical trial tables and find articles on “HCC” or “hepatocellular carcinoma”.

## 2. Etiology of HCC

There are many causes for the development of HCC. HBV and hepatitis C virus (HCV) are examples of viral infections that lead to the development of HCC [20]. Major risk factors of HCC change according to region where HBV is the dominant cause in high-risk areas while an increase in the number of people with cirrhosis is a risk factor in low-rate HCC areas [21]. Although HBV vaccination and viral hepatitis treatment programs have suppressed the occurrence of cirrhosis around the world, a rising burden of obesity, metabolic syndrome, and alcohol misuse are a threat to these trends [22] which may even overshadow these programs in the future [7]. Other influences of HCC development include metabolic syndrome-associated nonalcoholic fatty liver disease (MS-NAFLD), NASH, environmental factors such as heavy alcohol consumption or environmental toxins, and genetic illnesses [23]. Additionally, NAFLD is closely linked to type 2 diabetes mellitus (T2DM) and obesity as both T2DM, and obesity are prone to fat accumulation in hepatocytes [24]. Of the obese patients with T2DM, at least 90% have NAFLD [25] and insulin resistance does not help the NAFLD condition due to its association with lipid accumulation in hepatocytes, liver injury, and lipotoxicity [26]. NAFLD is also closely linked to dyslipidemia, which is a disorder that shows increased levels of triglyceride and low-density lipoprotein (LDL) and decreased levels of high-density lipoprotein (HDL) in the blood [27], and hypertension [28].

## 3. Progression of NAFLD to HCC

The illness of focus in this review is NAFLD and NASH, both of which are the leading cause of HCC and found in about 24% of the adult US population [23,28]. NAFLD describes a condition where lipids accumulate ectopically in the liver, not caused by increased alcohol consumption and other applicable reasons [29]. This accumulation of lipids describes steatosis. After steatosis, the severity of NAFLD depends on the development of necroinflammation and fibrosis, identified by the accumulation of collagen, an extracellular matrix proteins, and results in the formation of scar tissue [30]. If severe, the disease progresses to NASH and if cirrhosis also occurs at advanced fibrosis due to scarring and damage, the risk of developing cirrhotic HCC increases [29]. However, patients with non-cirrhotic NAFLD can also develop HCC [31] (Figure 1). Our lab has recently associated defective long-chain 3-hydroxy acyl-CoA dehydrogenase, an enzyme of fatty acid oxidation, as a new etiology of NAFLD–non-cirrhotic HCC [32]. Although the risk of NAFLD-associated HCC is low as compared to HBV and HCV related HCC, allowing NAFLD to progress to NASH increases the prevalence of HCC which is why there is a need for earlier surveillance, availability of non-invasive diagnostic and prognostic biomarkers and more treatment options for NAFLD and NASH [8,28]. A Markov model analysis even predicts that NAFLD-HCC in the US will increase by 122% in 2030 [33]. NAFLD is also difficult to treat, with the only treatment recommendation being lifestyle changes, likely because the mechanisms surrounding NAFLD are not completely understood, the tools used to image NAFLD are not cheap or completely accurate, and there are not enough non-invasive biomarkers [34].

## 4. Limitations of HCC Diagnosis and Treatment

Currently, there are limitations to how HCC is diagnosed. The current understanding of the pathogenesis of HCC, especially in the absence of liver cirrhosis, is poor [31]. There are minimal data that reinforce if HCC surveillance in NAFLD is useful or not, thus making it difficult to decide on the best surveillance methods [14]. Therefore, HCC is typically diagnosed at very advanced stages of NAFLD progression. In the last 15 years, there has not been much progress regarding surveillance/early detection, adjuvant therapies after resection or ablation, or supervision of intermediate stage HCC [15]. It is recommended that patients with cirrhosis are screened for HCC semi-annually, but screening usually occurs in less than 25% of patients at risk of HCC [35]. More importantly, NAFLD-HCC screening is difficult due to a low HCC incidence rate [35], therefore, there have been studies dedicated to finding molecular signatures to allow for better HCC risk prediction in NAFLD. Current surveillance may not be the best option either. Ultrasound is typically used for patients with cirrhosis, but the quality can be inadequate for HCC screening if patients are obese [36]. Ultrasound usage is also dependent on the operator and its sensitivity may vary depending on the severity of fibrosis [36,37]. Thus, attenuation imaging, which can detect hepatic steatosis through ultrasound imaging, may work better [38]. The non-cirrhotic version of NAFLD is even resistant to chemotherapy [15]. This is another reason why we chose to review clinical trials as there are clinical trials focused on surveillance and early detection while other clinical trials are focused on treatments or interventions to find a more permanent cure. So far, we did not come across any clinical trial focused solely on NAFLD–non-cirrhotic HCC.

## 5. Clinical Trials

NAFLD is reported in very heterogeneous populations with respect to its primary drivers and coexisting disease modifiers thereby resulting in hindrance to availability of highly potent drug treatments. The effective treatment of phenotypic manifestation of fatty liver disease requires that individual drivers should be targeted with accuracy based on patient’s phenotype and genetic background [3]. Currently, in the clinical trials, patients are recruited based on histologic grading and staging. However, this is problematic because same histologic phenotype is achieved due to malfunctioning of the combination of multiple pathways and thus predominant pathogenic pathway is not focused [39,40]. So, there is no across-the-board approach to deal with heterogeneity of NAFLD/NASH/HCC. Treatments and interventions used in clinical trials vary greatly, from altering one’s lifestyle to diagnostic testing and surveillance to surgical procedures and to drugs. We discuss all of these interventions in a sequential order. The stratification of patients in terms of age, sex, and severity of disease is included in the tables. All the trials are institutional review board-approved for all ethical issues and to ensure that scientific methods are adhered to in order to protect the rights and welfare of human research subjects.

### 5.1. Changes in Lifestyle (Diet, Exercise, and Naturopathy)

Lifestyle interventions focus on changing a patient’s lifestyle to improve health. Table 1 summarizes all the clinical trials related to lifestyle interventions. Amongst the seven clinical trials listed in the table, three are completed, while four are still recruiting or of unknown status. One of the lifestyle changes emphasizes weight loss because weight loss minimizes hepatic steatosis when paired with a hypocaloric diet and/or exercise, which further helps to alleviate liver injury [41,42,43]. Thus, these interventions exhibit how changes in diet or exercise affect NAFLD or how well naturopathy works in patients. Diets with caloric restriction and time-restricted feeding are some examples of lifestyle change [44]. A clinical trial focused on the calorie value of food items is NCT04861571, in which patients consume a very low-calorie diet (VLCD). The trial NCT04861571 specifically targets steatosis/fibrosis and uses transient elastography to measure mean tissue stiffness for hepatic fibrosis in patients [45]. Since advanced fibrosis in NAFLD patients increases the risk of developing HCC, transient elastography (TE) is a rapid, non-invasive technique to measure that risk factor.

The diet-based clinical trials also focus on the consumption of food supplements. One completed clinical trial (NCT04640324) uses nutraceutical treatment where participants with NAFLD carrying patatin-like phospholipase domain-containing protein 3 (PNPLA3)-rs738409, transmembrane 6 superfamily member 2 (TM6SF2)-rs58542926, or membrane-bound O-acetyltransferase 7 (MBOAT7)-rs641738 genetic variants consume 303 mg of silybin-phospholipids complex, 10 µg of vitamin D, and 15 mg vitamin E orally twice a day for six months. Amongst the 92-biopsy certified NAFLD patients, 30 were arranged as NAFLD wild type controls, 30 as wild type treated patients and 32 as mutated treated patients. The targets it focuses on are glycemia (FPG), insulinemia, homeostasis model assessment of insulin resistance (HOMA-IR), aspartate and alanine aminotransferases (ASTs and ALTs), C-reactive protein (CRP), thiobarbituric acid reactive substance (TBARS), stiffness, controlled attenuation parameter (CAP), dietary daily intake, and physical activity at baseline and end of treatment. HOMA-IR is used to determine if a patient has insulin resistance [48] and it can be used to predict hepatic fibrosis in patients with NAFLD [49]. AST and ALT are used to detect liver disease, and they are enzymes that normally resides in the cytosol of hepatocytes but are released in serum from liver cells when injury to the liver occurs [50]. In NAFLD, there is mild elevation in AST and ALT level [50], which can be linked to the development of HCC. After six months of nutraceutical treatment there is significant improvement in FPG, insulinemia, HOMA-IR, ALT, CRP, and TBARS (*p* < 0.05) in wild type-treated group, whereas no improvements were observed in the other two groups. Increased possibility of useful therapeutic outcome (*p* < 0.01) was reported in NAFLD wild type treated patients in terms of therapeutic regimen, independent of age, sex, comorbidities, medications, CAP, and stiffness in comparison to the mutated group. The clinical trial NCT04640324 concludes that PNPLA3, TM6SF2, and MBOAT7 play important roles in the development and worsening of NAFLD. The mutant variants of PNPLA3, TM6SF2, and MBOAT7 independently did not show any response to the silybin-based therapeutic plan and could be considered as useful predictive markers for non-responders [46]. The clinical trial NCT06031532 examined the effect of increased vitamin A and calcium intake with measurement of their serum levels, along with FibroScan and ultrasound results. Although this trial has been completed, no results have been posted yet.

Other diet related clinical trials are focused on using natural methods (naturopathy) as a form of treatment. The trial NCT03354247 focuses on type of food intake especially the mediterranean diet and its effect on patients’ physical activity level, body mass index (BMI), abdominal girth, and liver steatosis score; NCT03467282 examined the effects of probiotics supplementation on intestinal microbiota modulation, degree of hepatic fibrosis, inflammation, and body composition; and NCT03942822 examined the effect of adding milled chia seeds to the diet. NCT03942822 specifically targets the liver/spleen attenuation index which measures the difference between mean liver and spleen attenuation using Hounsfield Unit (HU) [51]. In computed tomography (CT) scans, HU is normally higher in the liver than the spleen [52]. If the HU attenuation is higher in the spleen than the liver, then the existence of liver fat may be diagnosed [52]. Chia seeds which are rich in omega-3 fatty acids like alpha-linolenic acid (ALA), antioxidants and fiber are used in the trial NCT03942822. Milled chia seeds (25 g/day) were given to 25 patients (30–70 years) and liver/spleen attenuation index along with visceral abdominal fat (VAF) were measured by computed tomography (CT) after two weeks of dietary stabilization (basal condition) and eight weeks of chia supplementation in the diet. Supplementation with chia seeds results in an increase in dietary fiber consumption (55%) and plasma alpha-linolenic acid (ALA) concentration (75%). As a result of this supplementation, a decrease in VAF (9%), body weight (1.4%), total cholesterol (2.5%), non-high-density lipoprotein cholesterol (3.2%), and circulating free fatty acids (FFA) (8%) was noticed with reverted or subsided NAFLD in 52% of the treated patients (*p* < 0.05 for all). The trial NCT03942822 is completed and concludes that chia seeds have the ability to avert metabolic abnormalities in NAFLD patients and thereby improve NAFLD condition [47]. Furthermore, this supplementation will help in improving the two most important complications of NAFLD: T2DM and coronary artery disease (CAD), as well as cirrhosis and HCC [53,54].

In addition to calorie value of food, food supplements, and natural diet alterations, exercise is included as another form of lifestyle intervention and recommended as it reduces the likelihood of developing NASH [44]. Exercise decreases the amount of hepatic fat by reducing oxidative stress, fatty acid synthesis, and inflammation while increasing fatty acid oxidation and levels of antioxidant enzymes [55,56]. The clinical trial NCT04835831 discerns the effects of adapted physical activity and dietetic advice on continuous CAP, anthropometric parameters (weight), and muscular performance.

Although diet alterations and weight loss exercises can avert metabolic abnormalities associated with NAFLD, long-term changes in lifestyle or weight loss are difficult to put into practice by the global NASH population. Only a small percentage of patients in general achieve this goal [43,57,58]. Therefore, NASH patients have need of a therapeutic approach to achieve long-term goals.

### 5.2. Diagnostic Screening and Surveillance

According to guidelines from many professional societies including AASLD, EASL, and Asian Pacific Association for the Study of the Liver (APASL), HCC surveillance was recommended only in at-risk individuals including patients with chronic HBV and cirrhosis from any etiology [59]. The challenges associated with HCC surveillance in NAFLD patients include increased difficulty recognizing relevant at-risk patients, HCC happening in absence of cirrhosis compared to other causes, cost-ineffectiveness of surveillance in non-cirrhotic patient population, poor performance of surveillance tools, and underutilization of HCC surveillance [14,60,61]. Several clinical trials focused on conducting diagnostic screening and surveillance as a form of intervention are in progress. Liver fibrosis is usually reversible in its early stages, and it can be used to predict mortality in NASH patients. Early diagnosis of liver fibrosis in patients is essential, therefore, many clinical trials are focused on screening [62,63]. Blood transaminase tests are commonly used to predict NAFLD progression but they are not reliable and sufficient [64]. Other biomarker tests being used to detect the risk of NAFLD progression into NASH and fibrosis include plasma cytokeratin 18 (CK18) fragment level [34]. However, CK18 fragment level, has limited accuracy and its sensitivity and specificity can be enhanced when combined with other biomarkers such as adiponectin, IL6, and resistin [65,66]. GALAD (Gender, age, AFP-L3, AFP, and des-carboxy-prothrombin [DCP]), a promising biomarker panel is proved to be excellent for early detection of HCC in NAFLD/NASH patients with and without HCC and GALAD is significantly superior to each individual marker [67]. Additionally, abnormal expression of non-coding RNAs, such as microRNAs (miRs), a new non-invasive biomarker, is used to detect NAFLD progression [68]. Deregulated levels of hepatic miRNAs, such as miR-122, miR-34a, and miR-29c; and circulating miR-192, miR-19, miR-125b, and miR-375 were noticed in NAFLD patients with a positive correlation to disease severity [69,70,71]. The most noticed miRNA is miR-122, which is usually found in great abundance in the liver, making it a biomarker contender [71].

Diagnostic tools are essential for prevention and diagnosis of disease severity. Ultrasound, the most used diagnostic tool, has its own limitations in diagnosing NAFLD. Although magnetic response spectroscopy (MRS) is non-invasive and quantifies fat content of the liver, magnetic resonance imaging-derived proton density fat fraction (MRI-PDFF) is the best technique used to evaluate fat content in the liver. TE examines liver fibrosis with FibroScan and an M probe, whereas magnetic resonance elastography (MRE) is an accurate, non-invasive tool that examines liver stiffness and fibrosis stage [34]. Fourteen clinical trials are listed in Table 2, which belong to the diagnostic screening and surveillance category. Out of these fourteen trials, only one clinical trial is completed (NCT04834063) while others are of unknown status, actively recruiting or have not started recruiting yet. Although the involvement of macronutrients in NAFLD pathogenesis is well established, the contribution of micronutrients in NAFLD pathogenesis has gathered very less attention. The observational clinical trial NCT04834063, is designed to include two groups: experimental group including 70 patients diagnosed with NAFLD on ultrasound exam and control group of 30 healthy volunteers on ultrasound exam. The level of micronutrient, zinc, and selenium in serum are measured, along with FibroScan measurement, to examine their association with hepatic fibrosis. Despite the completion of this trial, no conclusive results were posted. A behavioral study being conducted at different time intervals to examine liver cancer risk in NAFLD and HCC patients (NCT05870969), is still recruiting patients. NCT05802199 is focused on testing hepatic fat and hepatic fibrosis so that the efficiency of ultrasound derived fat fraction (UDFF) for hepatic steatosis can be compared to MRI-PDFF. The clinical trial NCT05754385 is targeted towards ambulatory monitoring of liver fat and standard of care to ultimately find out the percentage of participants with significant change in hepatic fat. NCT06101758 uses the handgrip strength test, total body dual energy X-ray absorptiometry, and muscle ultrasound to examine the prevalence of sarcopenia in participants. Another trial NCT03811236 applies cold exposure intervention with measurement of hepatic lipid content (%) over six weeks using MRI-PDFF and the volume of brown adipose tissue by 18-fluorodeoxyglucose positron emission tomography/MRI in NAFLD patients. In addition, the clinical trial NCT04820036 pushes many interventions—insulin resistance, quality-of-life assessment, liver function test, radiologic and serologic features of NASH, endoscopic ultrasound (EUS), and endoscopic sleeve gastroplasty—to measure EUS-guided liver biopsy with portal pressure gradient from zero to 12 months, while NCT05486429 uses the abdominal ultrasound and lipid profile to measure serum total cholesterol, triglyceride, HDL, LDL, and VLDL.

Meanwhile, two clinical trials use blood collection as a form of intervention. One requires blood collection to assess liver stages through biochemical, genetic, and immunocytochemistry methods (NCT03307408) while the other examines methylation patterns based on molecular blood tests (NCT04264754). For many cancers, the methylation pattern serves as a biomarker, as DNA methylation of tumor suppressor genes occurs early and differently in patients with/without HCC [72]. Unfortunately, the clinical trial based on blood methylation pattern NCT04264754 is terminated without any conclusive explanation.

The clinical trial NCT05370053 depends on the enhanced liver fibrosis (ELF) test to target F3 to F4 fibrosis stages over two years. The ELF test measures hyaluronic acid, procollagen III amino acid terminal peptide, and tissue inhibitor of metalloproteinase in the blood which are all molecules that can be found in the liver matrix metabolism [73]. The score for the ELF test indicates the severity of liver fibrosis in a patient. F3 and F4 are stages for liver fibrosis, ranging from absent (F0) to cirrhosis (F4), and F3 to F4 progression is considered to be at the advanced fibrosis stage [74]. In clinical trial NCT04785937, ultrasound and magnetic resonance are being used to target false positives and negatives, sensitivity, and specificity of each imaging parameter in comparison to liver biopsy and the ability to diagnose NASH or fibrosis. Further, two trials which are active but have not started recruiting yet are NCT05165446 and NCT06328283. The trial NCT05165446 will use image-based surveillance to perform tissue viscoelasticity assessments through novel MRE and NCT06328283 cites use of glutamine synthetase and BCL2 associated transcription factor 1 (BCLAF1) as a form of intervention and targets their measurement in the control (cirrhotic patients with no HCC) and experimental group (cirrhotic patients with early HCC).

In conclusion, the guidelines for surveillance by different professional societies, as well as efficacy of different screening methods used in the clinical trials, should be strictly updated regularly.

### 5.3. Surgical Procedures

Surgical procedures may be used as a form of intervention. One example is bariatric surgery, which provides a fast way to lose weight for patients with NAFLD/NASH as compared to diet and exercise [44]. Bariatric surgery performed on morbidly obese patients can also alleviate steatosis, NASH, and liver fibrosis in 30% of patients, but it comes with risks, so it is not usually chosen as a first-line treatment [75]. Six clinical trials of this category are listed in Table 3, and they use surgical methods to treat patients with NASH. Three of them are completed (NCT03536650, NCT03524365, and NCT04677101) while the other three are of unknown status (NCT04281303 and NCT04653311) or not yet recruiting (NCT05623150). Amongst the completed clinical trials, NCT03536650 evaluates the effects of novel minimally invasive endoscopic approach of duodenal mucosal resurfacing (DMR) in NASH patients to look at adverse device effects over one year with no results posted so far, while NCT03524365 (BRAVES) uses Rouxen-Y gastric bypass (RYGB) or sleeve gastrectomy (SG), along with intensive lifestyle modification (restricted diet, gradually increased walking, and moderate physical activity), to see a histological resolution of NASH without the worsening of fibrosis after a year of intervention [76]. The clinical trial NCT04677101 (LIBRA) uses liver biopsy to assess blood biomarker sensitivity and accuracy in participants to see if they can be predictive of fibrosis stage and NASH. The multicenter randomized clinical trial NCT03524365 (BRAVES) enrolled 100 subjects in the age range of 46.9 ± 10.5 years (67% women) as the discovery cohort, whereas NCT04677101 (LIBRA) enrolled 150 subjects aged 43.4 ± 11.9 years (56% women) as the validation cohort. In this observational study (NCT04677101), liquid biopsy, which is a less invasive form of liver biopsy, is used as an alternative method. Liquid biopsy also conveys several unfulfilled clinical needs in terms of sensitivity, specificity, prognosis, and prediction of therapeutic response [76]. Both clinical trials NCT03524365 (BRAVES) and NCT04677101 (LIBRA) tested perilipin-2 (PLIN2) and a member of RAS oncogene family (RAB14) in circulating monocytes as a prognosticator of histological NASH and liver fibrosis, respectively. The plasma biomarkers used so far have low sensitivity (from 62% to 66%), and specificity (from 78% to 82%) [66,77,78,79]. However, accuracy, sensitivity, and specificity for PLIN2 were 93%, 95%, and 90% in the discovery cohort and 92%, 88%, and 100% in the validation cohort, respectively. Similarly, accuracy, sensitivity, and specificity for RAB14 were 99.25%, 100%, and 95.8% in the discovery cohort and 97.6%, 99%, and 89.6% in the validation cohort, respectively. The area under the receiver operating characteristic (AUROC) for NAFLD activity score (NAS) ranged from 83.7% in the discovery cohort to 97.8% in the validation cohort [76]. Thus, both BRAVES and LIBRA conclude that PLIN2 and RAB14 in circulating monocytes are reliable non-invasive biomarkers of NASH and fibrosis, respectively. Moreover, they can replace invasive liver biopsy-based histology for diagnosis and management of NASH and liver fibrosis. Measuring PLIN2 and RAB14 in a single blood test is inexpensive, and hundreds of samples can be analyzed in a day. Also noticed was the fact that fresh blood is as good as cryopreserved blood for monocyte flow cytometry. Unfortunately, in BRAVES and LIBRA clinical trials, only Caucasian subjects were enrolled, thus restricting the findings to be applied, in general, to other ethnicities [76].

Another clinical trial using liver biopsy as an intervention is NCT05623150, which will assess variations in gene markers, metabolic gene markers, epigenetic markers, and tissue proteins. Clinical trial NCT05623150 is going to last 10 years and has not started recruiting yet. The study will measure mRNA variations in metabolic gene markers involved in inflammation and/or metabolism in patients with and without HCC and having different levels of severity of liver damage. Furthermore, variations in gene markers in cell death, regeneration, tumors and single-nucleotide polymorphisms (SNPs), and epigenetic markers (histone methylation and miRNA) will also be compared in the same set of patients. Finally, variations in tissue proteins between tumor and non-tumor tissues based on proteomics, as well as radiomics, will also be compared.

The current non-invasive technique with less complications and similarity in results to traditional invasive surgical methods is endoscopic vertical gastroplasty (EVG). Endoscopic vertical gastroplasty has never been evaluated in obese and NASH/cirrhosis patients. NCT04281303 clinical trial uses EVG to evaluate obese participants with NASH/cirrhosis. The study is planned to assess the safety and efficacy of EVG in adult populations with obesity and NASH/cirrhosis in improving metabolic factors, liver histology, and one stage in fibrosis without worsening of NASH. The clinical trials’ targets include looking for adverse events that may arise due to the treatment used, for example, the model for end-stage liver disease (MELD) and the Child–Pugh score. The MELD score helps predict the survival of patients with advanced liver disease [80] and the Child–Pugh score acts as a prognostic indicator in patients with cirrhosis [81]. The last clinical trial listed in Table 3 uses endoscopic sutured gastroplasty, along with an endomina device, to evaluate the rate of NASH disappearance without any worsening of the fibrosis grade (NCT04653311). Aside from these clinical trials, another treatment procedure for NAFLD/NASH could be fecal microbiota transplantation (FMT). In FMT, a healthy donor’s fecal bacteria are transplanted into a patient to repopulate the gut microbiome [44]. However, no such trial has been designed yet.

In conclusion, less expensive non-invasive surgical techniques with results comparable to invasive techniques would be a better option for NAFLD/NASH treatment. More clinical trials should be conducted to check accuracy, sensitivity, and specificity of these alternative non-invasive surgical procedures.

### 5.4. Drug Intervention

Many clinical trials test drugs as a form of treatment because there is no official drug treatment for NAFLD. In general, many potential drugs used in treatment act upon molecules associated with hepatic metabolism, hepatocyte apoptosis, inflammatory pathways, the extracellular matrix, and hepatic stellate cell (HSC) activation [26]. Drugs may also result in metabolic disruption of the liver by affecting lipotoxicity, oxidative stress, mitochondrial dysfunction, and fibrosis [34]. The drugs examined in recent clinical trials are empagliflozin, curcuvail, obeticholic acid, dapagliflozin, pioglitazone, lisinopril, tislelizumab, hydroxychloroquine, DNP007, lenvatinib, sorafenib, MGL-3196 (resmetirom), and emricasan (NCT05694923, NCT06256926, NCT03439254, NCT05254626, NCT04501406, NCT04550481, NCT05622071, NCT05733897, NCT06400771, NCT05391867, NCT03900429, NCT05500222, and NCT03479125) (Table 4). Amongst the thirteen clinical trials, four of them are completed with or without results being posted on clinicaltrials.gov (NCT05694923, NCT06256926, NCT03439254, NCT05254626). Empagliflozin is a sodium glucose co-transporter type-2 inhibitor (SGLT-2i) that regulates the events implicated in NAFLD pathogenesis which are low-grade inflammation, endoplasmic reticulum (ER) and oxidative stress, autophagy, and apoptosis [82]. In a randomized trial NCT05694923, patients receive either 10 mg of empagliflozin or interventions like diet control. Hepatic steatosis and fibrosis are targeted to assess CAP score over three months. The CAP score helps detect and grade hepatic steatosis through ultrasonic signals sent by a FibroScan device [83]. NCT06256926 is a randomized, double-blind, placebo-controlled trial, designed to compare efficacy, safety, and tolerability of 250 mg curcuvail capsule (*Curcuma longa* extract containing 35% curcuminoids) consumed orally twice daily by NAFLD patients over 60 days. With curcuvail, the primary target to be examined is the change in NAFLD grading (grade 0 to grade 3), based on a liver ultrasound from baseline to day 60. Grade 0 of NAFLD grading is no fat accumulation, while grades 1–3 are mild, moderate, and severe increases in echogenicity, respectively. Also, in grade 1–3, there is normal, slightly, to severely compromised visualization of diaphragm, intra-hepatic vessel borders and posterior portion of the right hepatic lobe. The secondary targets to be measured include AST-to-platelet ratio index (APRI); fibrosis score; CAP; changes in lipid profile, as well as liver enzymes (ALT and AST) from baseline to 60 days; and adverse or seriously adverse events till 67 days from day of randomization. The clinical trial NCT03439254 (REVERSE), uses obeticholic acid (OCA) to improve fibrosis histologically with no worsening of NASH in adults with NASH/cirrhosis. OCA is a semi-synthetic bile acid derivative which is a farnesoid X receptor (FXR) ligand. FXR, a nuclear receptor, regulates several processes in the liver including inflammation, fibrosis, and metabolism of bile acids and glucose [84]. OCA by activation of FXR in hepatocytes and enterocytes leads to a decrease in bile acid synthesis and improvement of inflammation and hepatic steatosis [85,86]. In this phase 3 trial, randomized adults with compensated cirrhosis due to NASH consumed a 10 or 25 mg obeticholic acid tablet or placebo orally once daily for up to 18 months and the efficacy and safety of obeticholic acid was evaluated in terms of improvement in fibrosis by at least one stage without worsening of NASH. The trial NCT03439254 did not satisfy its primary endpoints (improve fibrosis histologically with no worsening of NASH) by 18 months and therefore halted at that point. The failure of meeting endpoints is probably because of lack of cirrhosis sub stratification just like other trials involving cirrhotic patients. A global, multicenter, phase3 randomized clinical trial, NCT02548351 (REGENARATE), involving OCA treatment to improve NASH with fibrosis is also terminated (September 2015 to September 2023) and not included in the table because of its start date in 2015 [87]. Dapagliflozin and pioglitazone are used to target the NAS in clinical trial NCT05254626. Dapagliflozin is a SGLT-2i, while pioglitazone is a peroxisome proliferator-associated receptor-γ (PPAR-γ) agonist. The efficacy and safety of dapagliflozin (10 mg) as compared to pioglitazone (30 mg) in diabetic and non-diabetic patients with NASH was assessed in terms of NAS. The NAS is calculated on a scale from 0 to 8 using scores of 0 to 3 for steatosis, 0 to 3 for lobular inflammation, and 0 to 2 for hepatocyte ballooning [88]. The NAS helps in identifying if NAFLD patients have developed NASH, which is highly likely if the score is ≥5 and not likely if the score is ≤3. Pioglitazone is used to treat type 2 diabetes, but it has been used in trials to treat non-diabetic NASH patients, yielding positive results; however, it should not be used for patients with high risk or established heart failure (NCT00063622—not included in Table 4) [89,90].

In NCT04501406, which is a two-arm, randomized, double-blind, placebo-controlled trial, the effect of 15 mg of pioglitazone treatment every day for 72 weeks on liver histology in NASH patients was examined. An improvement in NAS by ≥2 points without an increase in fibrosis stage (NCT04501406) was considered the target to be achieved. Regardless, pioglitazone can only be used to treat patients with NAFLD if they have biopsy-proven NASH based on 2018 AASLD practice guidelines [92]. The clinical trial NCT04550481 is designed to study the role of lisinopril in preventing progression of NAFLD. Patients receive lisinopril orally once daily for 24 weeks in absence of unacceptable toxicity, undergo transient elastography and blood sample collection and may undergo PDFF MRI and MRE and follow-up at 32 weeks. Lisinopril was used mainly to target N-terminal type III collagen propeptide (PRO-C3) and cross-linked multimeric PRO-C3 (PC3X). PRO-C3 and PC3X are biomarkers for liver fibrosis and HCC progression. PRO-C3’s role in HCC is unclear because it cannot discern between cross-linked N-terminal pro-peptides and those that are single stranded [93]. However, both PRO-C3 and PC3X can be measured to differentiate between patients with and without HCC. Other targets of NCT04550481 are steatosis, CAP, NAFLD fibrosis score (NFS), and inflammatory markers such as caspase cleaved CK-18, NF-kB, TGF-b, TNF-a, IL6, and IL8. The NFS predicts the presence or absence of advanced fibrosis based on many criteria including age, AST/ALT ratio, platelet count, hyperglycemia, albumin level, and BMI [74].

A few other clinical trials use tislelizumab, lenvatinib, and sorafenib to target clinical endpoints (NCT05622071 and NCT05391867). These clinical endpoints include objective response rate (ORR), progression-free survival (PFS), and overall survival (OS). ORR measures how the treatment of interest affects tumor burden in a patient known to have or have had solid tumors [94]. PFS is the time of randomization to the first sign of disease progression or death, while OS is the time of randomization to death [94]. Tislelizumab also targets the frequency of limiting toxicity, defined as any adverse event related to the drug resulting in discontinuation of treatment before the second dose. In an ongoing clinical trial, NCT05622071, 200 mg of tislelizumab will be administered intravenously (IV) every 3 weeks, and treatment should be continued for a maximum two years. It is an anti-PD1 agent, and its efficacy is tested in terms of ORR, OS, and PFS in patients with HCC and moderately altered liver functions. However, in NCT05391867, lenvatinib and sorafenib were assessed against each other for their efficacy in the management of advanced HCC in terms of OS. One group of patients received 8 mg/day of lenvatinib capsules in two divided doses, while the other group of patients received 400 mg/day of sorafenib tablets in two divided doses. The liver was examined by CT or MRI using a triphasic scanning technique, and tumor assessments were performed using liver function tests in patients every six weeks. The study is designed to measure OS from the date of randomization up to three months or till death—whichever is earlier.

A significant decrease in ALT levels will be examined in recruited patients before and after hydroxychloroquine (HCQ) treatment over one year in the ongoing clinical trial NCT05733897, while the safety, tolerability, pharmacokinetics, and immunogenicity of DNP007 will be evaluated in NCT06400771. The trial NCT06400771 is a phase 1 study in continuation for exploratory evaluation of single IV dose of DNP007 (1–8 mg/kg) over 30 min and will start with a minimum number of subjects (12 people). The safety and pharmacokinetic studies will be performed after single IV administration for up to one month. Another clinical trial, NCT03479125, is a post-treatment follow-up study and uses ultrasound, along with MRI and CT on patients who previously received at least one dose of emricasan or placebo to examine the adjusted event rate for HCC. However, this trial was terminated because the production of emricasan by the sponsor was discontinued.

Recently, the Food and Drug Administration (FDA) has approved an oral liver-directed drug, MGL-3196 (resmetirom), to treat NAFLD-NASH (FDA, 14 March 2024). It is a thyroid hormone receptor beta (THR-β)-selective agonist that leads to decreased intrahepatic lipid content through its control on hepatic triglyceride and cholesterol metabolism [95]. Resmetirom affects patients’ lipid profiles by reducing blood levels of atherogenic lipids; thereby, it may reduce the incidence of cardiovascular disease, which is the primary cause of mortality in NASH patients [96]. Prior to this approval, there was no treatment available for NAFLD-NASH besides lifestyle changes. Hence, the FDA created an accelerated pathway for clinical trials if they could improve the liver fibrosis stage or achieve a resolution of NASH for clinical benefit and reduction of clinical outcomes [91]. As two primary histologic endpoints, the clinical trial NCT03900429 for resmetirom examined if usage of the drug resulted in NASH resolution without any worsening of fibrosis and a two-point or more reduction in NAS score over 52 weeks. In this phase 3 clinical trial, participants with biopsy-confirmed NASH and a fibrosis score of F1B to F4 (F0:no fibrosis to F4: cirrhosis) were randomized (1:1:1) to take placebo, 80 of resmetirom, or 100 mg of resmetirom, and liver biopsy was performed. It was noticed that, after 52 weeks, both doses of resmetirom worked better than the placebo, fibrosis did not worsen in patients who took resmetirom, and a higher percentage of patients showed an improvement in fibrosis by at least one stage, along with a lack of worsening of NAS in comparison to the placebo [91]. Improvement in NASH with no augmentation of fibrosis was accomplished in 25.9%, 29.9%, and 9.7% patients in the 80 mg resmetirom, 100 mg resmetirom, and placebo group, respectively, at *p* < 0.001. Similarly, fibrosis was improved by at least one stage, with no worsening of the NAS in 24.2% (80 mg resmetirom), 25.9% (100 mg resmetirom), and 14.2% (placebo group) patients at *p* < 0.001. The investigators also found that LDL levels were reduced and noted an improvement in levels of triglycerides, HDL, cholesterol, and other lipids and lipoproteins from baseline, as well as reduced lipid enzyme levels [91]. The LDL levels were reduced in 13.6% (80 mg resmetirom), 16.3% (100 mg resmetirom), and 0.1% (placebo group) patients from baseline to week 24 at *p* < 0.001. In terms of safety, patients commonly experienced gastrointestinal problems (diarrhea and/or nausea). The incidence of adverse events was similar across the trial groups, with 10.9% in the 80 mg resmetirom group, 12.7% in 100 mg resmetirom group, and 11.5% in the placebo group [91]. However, more patients from the group taking 100 mg resmetirom chose to discontinue the trial due to adverse events. In conclusion, both doses of resmetirom were found to be effective with reference to the two primary histologic endpoints in patients with NASH and liver fibrosis. The major limitation of NCT03900429 is the lack of correlation of histologic data with clinical outcomes. The safe use of resmetirom for extended period is also not assessed yet. The trial is designed to continue till 54 months to follow and assess liver-related outcomes, including progression to cirrhosis.

Another ongoing clinical trial with resmetirom is NCT05500222, which is on clinical outcomes in patients with well-compensated NASH cirrhosis, and it is still recruiting patients. This is a multi-national, multicenter, double-blind, randomized, placebo-controlled study. Randomized participants 3:1 will receive 80 mg resmetirom or placebo once daily in the morning for the duration of trial or until the required number of composite clinical outcome events are achieved. Patients will be monitored for progression to a composite clinical outcome event such as liver decompensation events (ascites, hepatic encephalopathy, or gastroesophageal variceal hemorrhage) and confirmed increase in MELD score from <12 to ≥15. The study is meant to be a 60-day screening period with a 3-year treatment period.

### 5.5. Combination Drug Therapy

Based on heterogeneity of NAFLD/NASH, no one drug is effective for all patients. Combination therapies, including more than one drug from a rich pipeline of compounds, may be used to take care of NAFLD/NASH heterogeneity with possibly many drivers for its pathogenesis. The combination therapies, targeting multiple pathogenic mechanism, could increase the efficacy of individual therapies or reduce side effects of individual therapies in NASH, especially in patients with advanced fibrosis. Combination therapies will permit a personalized and ideal patient care approach in NAFLD/NASH. Six completed clinical trials related to combination therapy are summarized in Table 5. One of the completed clinical trials, in the category of increasing efficacy, is NCT03449446 (ATLAS); it assessed the safety and tolerability of selonsertib (SEL), firsocostat (FIR), and cilofexor (CILO) and evaluated the liver fibrosis changes without worsening of NASH. In this multicenter, multinational, phase2b trial (NCT03449446), 392 patients in F3/F4 stage of liver fibrosis (bridging fibrosis or compensated cirrhosis) were randomized to receive placebo, SEL 18 mg, CILO 30 mg, or FIR 20 mg once daily for 48 weeks, alone or in a combination of two drugs. The primary endpoint is improvement by ≥1 stage in fibrosis without worsening of NASH from start to 48 weeks, and exploratory endpoints include changes in NAS, machine learning (ML)-based changes in liver histology, biochemistry of liver, and non-invasive markers. Most of the patients had cirrhosis (56%) and NAS ≥ 5 (83%). The primary endpoint was attained in 11% placebo treated patients versus CILO/FIR (21%; *p* = 0.17), CILO/SEL (19%; *p* = 0.26), FIR/SEL (15%; *p* = 0.62), FIR (12%; *p* = 0.94), and CILO (12%; *p* = 0.96). CILO/FIR resulted in a significant decrease in liver fibrosis score (*p* = 0.040) and a shift in fibrosis pattern from bridging fibrosis to ≤F2. A higher proportion of patients receiving CILO/FIR as compared to placebo had ≥2 point reduction in NAS, as made evident by a reduction in steatosis, lobular inflammation, and ballooning and significantly improved AST, ALT, CK18, ELF score, and liver stiffness (all *p* ≤ 0.05). An increase in both LDL cholesterol and triglycerides was also observed. Although NCT03449446 did not meet the primary endpoint (less statistical significance), the researchers indicated that CILO/FIR combination can possibly be used for the long-term therapy of patients with bridging fibrosis due to NASH [97]. Further studies are in need to confirm efficacy and safety of CILO/FIR and to determine if there is any reduction in clinically relevant outcomes, such as hepatic decompensation and mortality.

The clinical trial NCT03517540 (TANDEM) is a randomized, double-blind, multicenter study to assess the safety, tolerability, and efficacy of tropifexor (TXR) and cenicriviroc (CVC) combination therapy in adult patients with NASH and fibrosis stage F2/F3 [98]. TXR is a farnesoid X receptor agonist, whereas CVC is a chemokine receptor type 2/5 (CCR2 and CCR5) antagonist, and both target the pathways involved in NASH, i.e., steatotic, inflammatory, and/or fibrotic pathways. Two hundred patients were randomized in a 1:1:1:1 (50:50:50:50) ratio to receive once daily either 140 µg TXR, 150 mg CVC, TXR 140 µg + CVC 150 mg, or TXR 90 µg + CVC 150 mg. The study aims for a 48-week treatment with a 4-week follow-up, and efficacy is assessed by ≥1-point improvement in liver fibrosis score versus baseline or resolution of NASH after 48 weeks. The acceptable safety and tolerability dose of TXR and CVC is 60 µg and 150 mg, respectively, from phase 1 DDI study. The ongoing FLIGHT-FXR study suggested a higher dose of TXR (140 µg) in the TANDEM trial, which is expected to reach a near-maximal response for multiple biomarkers. The individual monotherapies are the comparator arms instead of placebo. FXR agonists are proven therapies for NASH, including the bile acid FXR agonist OCA (FLINT and REGENERATE) [100]. However, OCA is related to pruritus and increased lipid levels, and its steroidal backbone contributes to low aqueous solubility and bioavailability. Therefore, the non-bile, non-steroidal FXR agonist TXR is used in the trial TANDEM. TXR reduces synthesis of triglycerides by inhibiting SREBP1C in the hepatocytes by activating FXR and creating an antisteatotic effect [101]. TXR also manifests anti-inflammatory activity in the hepatocytes through upregulation of SHP and suppression of NF-κB [102] and downregulation of fibrogenic markers Col1a1 and TIMP1 in HSCs, resulting in an antifibrotic effect [103]. CVC, on the other hand, inhibits CCR2 and CCR5, resulting in a reduction in inflammatory signals. Altogether, CVC shows anti-inflammatory and antifibrotic effects [104]. The investigators of the TANDEM trial hope to develop new combination therapy for NASH treatment. Another clinical trial, NCT04065841 (ELIVATE), evaluating combination therapy using TXR and licogliflozin (a sodium-glucose cotransporter 2 inhibitor) was terminated, while NCT03987074, assessing semaglutide, FIR, and CILO; and NCT05415722, assessing TERN-501 and TERN-101, were completed. TERN-501, a THR-β agonist, and TERN-101, an FXR-agonist, are used alone or in combination to reduce hepatic fat content in patients with non-cirrhotic NASH.

FXR agonists such as OCA, CILO, and FIR adversely affect the cardiovascular system usually because they result in hypertriglyceridemia. Cardiovascular disease is the leading cause of death in NASH patients. Therefore, combination therapies to reduce the side effects of FXR agonists were tried in clinical trials. One such trial is the phase 2 NCT02781584, in which safety and efficacy of fenofibrate and icosapent ethyl (Vascepa) to mitigate elevated levels of triglyceride (≥150 and <500 mg/dL) in NASH/NAFLD patients previously treated with CILO (FXR agonist) and FIR (aceyl-CoA carboxylase inhibitor) was assessed. Patients (n = 30) with elevated triglycerides were randomized (n = 30) to receive Vascepa 2g twice daily or fenofibrate 145 mg daily for two weeks, followed by addition of CILO 30 mg and FIR 20 mg daily for 6 weeks. The targets used to assess were safety, lipids, and liver biochemistry. Investigators noted that all treatments were well tolerated, adverse events were of mild-to-moderate severity (grade 1 to 2), and no discontinuation of therapy was observed. In both the Vascepa and fenofibrate groups, the median triglyceride levels were similar (177 and 190, respectively). Median changes in triglycerides from baseline for Vascepa vs. fenofibrate after 2 weeks of pretreatment were -12 mg/dL (*p* = 0.09) vs. −32 mg/dL (*p* = 0.012), and after 6 weeks, they were +41 mg/dL (*p* = 0.001) vs. −2 mg/dL (*p* = 0.92). In patients with baseline triglycerides <250 mg/dL, fenofibrate was more effective as compared to Vascepa in mitigating triglycerides elevated after 6 weeks of combination treatment (+6 vs. 39 mg/dL). Similar results were observed in patients with baseline triglycerides ≥250 mg/dL (−61 vs. +99 mg/dL, respectively). The conclusion drawn from the trial is that fenofibrate, but not Vascepa, initiated prior to CILO and FIR, was safe and more effective in mitigating increased triglycerides associated with FIR, thereby reducing the adverse effect on cardiovascular system [99].

Based on heterogeneity of NAFLD/NASH, no one drug is effective for all patients, so having a rich pipeline of drugs and combination therapy will permit personalized and ideal patient care.

## 6. Conclusions and Future Direction

The prevalence of NAFLD-NASH-HCC is growing globally, leading to the continuous updating of clinical guidelines. NAFLD, NASH, and HCC are difficult to treat because of their late detection and limited surveillance. The only treatment recommendation for NAFLD is lifestyle changes, most likely because the mechanisms involved with NAFLD are not completely understood. Each intervention used in the clinical trials has its own limitations. The current imaging tools used in diagnostic screening and prevention have limited sensitivity and accuracy, they are too expensive for patients, or diagnostic surgical procedures are invasive and carry complication risks. There are not enough non-invasive biomarkers or molecular signatures available to allow for better HCC risk prediction in NAFLD. The long-term prognosis of patients with HCC after surgery is still far from satisfactory. Oral therapy with lenvatinib can somehow reduce postoperative recurrence and improve long-term survival of patients. As for drugs, only resmetirom has been approved to treat NAFLD-NASH, while many other drugs are still being tested.

Most of the listed trials are either currently active and running or have not started recruiting yet. With the completion of these trials, we may, in the future, find more defined lifestyle options, better surveillance and biomarkers or molecular signatures, more sensitive non-invasive surgical procedures, and the availability of more drugs to treat NAFLD-NASH to save the global population from getting HCC or lead to its early detection. In the future, platform trials (adaptive master protocol) are needed to overcome major limitations of traditional clinical trials. They are more efficient, uniform clinical trials with an emerging field of new compounds, providing quicker, consistent evidence and allowing faster transition between trial phases. The rich pipeline of new drugs and compounds and use of combination drug therapy will empower the personalized and ideal patient care to account for NAFLD/NASH heterogeneity. These will benefit the various stakeholders, for example, sponsors, IMP owners, and patients.

## Figures and Tables

**Figure 1 ijms-26-00306-f001:**
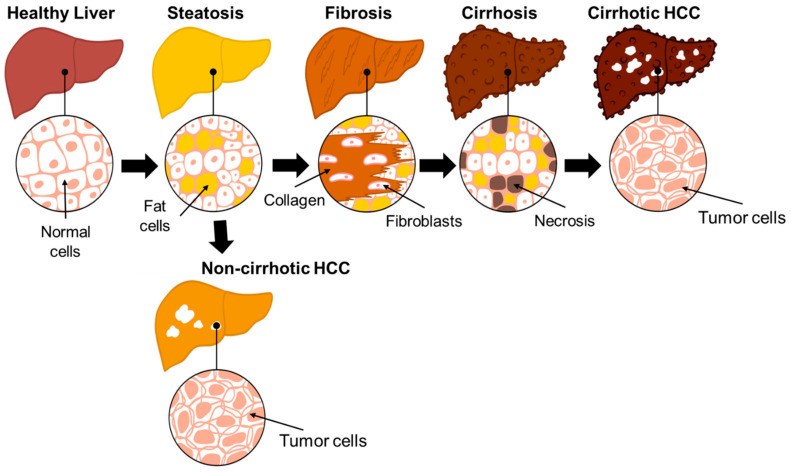
Disease progression of a healthy liver to HCC. Stages of progression of a healthy liver to HCC: The liver first experiences steatosis where fat cells become abundant, followed by fibrosis, where fibroblasts and collagen form large amounts of scar tissue; and ultimately, cirrhosis, where necrosis appears in addition to the fat cells and scar tissue, leading to cirrhotic HCC, where tumor cells are formed. Non-cirrhotic HCC can also occur after steatosis with the formation of tumor cells.

**Table 1 ijms-26-00306-t001:** Clinical trials with changes in lifestyle (diet, exercise, and naturopathy).

Clinical Trial #	# of Participants	Eligible Age/Sex	Intervention	Target	Conditions	Phase or Study Type/Status	Start and End Date	References
NCT04861571	20	18–70 years old/all	Very low-calorie diet	Steatosis/transient elastography	NAFLD	Phase 1/ recruiting	Oct. 2023 to Dec. 2025	Trial contd.
NCT04640324	92	18–80 years old/all	Nutraceutical therapy	HOMA-IR over 6 months/ALT	NAFLD and insulin resistance	Not applicable (NA)/completed	Jan. 2017 to Apr. 2018	[46]
NCT06031532	110	20–80 years old/all	Vitamin A and Calcium	Serum vitamin A and calcium level, FibroScan, and ultrasound	NAFLD	Observational/completed	May. 2023 to Aug. 2023	No results posted
NCT03354247	100	18–75 years old/all	Lifestyle	Changes in food intake and Mediterranean diet adherence score, physical activity level, BMI, abdominal girth, and liver steatosis score	NAFLD	NA/unknown	Jul. 2017 to Dec. 2020	No results posted
NCT03467282	46	18–80 years old/all	Probiotic	Hepatic fibrosis and cardiovascular risk	NAFLD	NA/unknown	Nov. 2017 to Dec. 2021	No results posted
NCT03942822	25	30–70 years old/all	Milled chia seeds	Liver/spleen attenuation index	NAFLD	NA/completed	Sep. 2016 to Sep. 2017	[47]
NCT04835831	105	18–90 years old/all	Adapted physical activity and dietetic advice	Continuous CAP decreased by 10%	NAFLD	NA/recruiting	Sep. 2021 to Apr. 2026	Trial contd.

**Table 2 ijms-26-00306-t002:** Clinical trials with diagnostic testing and surveillance.

Clinical Trial #	# of Participants	Eligible Age/Sex	Intervention	Target	Conditions	Phase or Study Type/Status	Start and End Date	References
NCT04834063	80	20–80 years/all	Serum zinc and selenium level and FibroScan measurement	Serum zinc and selenium levels and their association with hepatic fibrosis	NAFLD	Observational/completed	Mar. 2021 to Sep. 2021	No results posted
NCT05870969	20000	18–75 years/all	Behavioral/liver cancer surveillance	Three-month, six-month, and yearly surveillance of liver cancer	Hep B, Hep C, HCC/Cirrhosis, and NAFLD	Observational/recruiting	Mar. 2023 to Mar. 2028	Trial contd.
NCT05802199	300	18 years and older/all	Hepatic fat and hepatic fibrosis	The efficiency of UDFF (in %) for hepatic steatosis in comparison with MRI-PDFF (in %).	NAFLD	NA/recruiting	Jan. 2023 to Dec. 2023	No results posted
NCT05754385	260	18–65 years/all	Ambulatory monitoring of liver fat and standard of care	Percentage of subjects with significant change in hepatic fat	NAFLD	NA/recruiting	May. 2023 to Oct. 2025	Trial contd.
NCT06101758	125	18–69 years/all	Handgrip strength test, total body dual energy X-ray absorptiometry, and muscle ultrasound	Prevalence of sarcopenia in patients	NAFLD, liver cirrhosis, and HCC	NA/recruiting	Oct. 2023 to Jul. 2027	Trial contd.
NCT03811236	26	18–50 years/all	Cold exposure	Hepatic lipid content (%) over six weeks by magnetic resonance imaging-proton density and brown adipose tissue volume over six weeks by 18-fluorodeoxyglucose positron emission tomography/magnetic resonance imaging	NAFLD	NA/recruiting	Jan. 2019 to Dec. 2026	Trial contd.
NCT04820036	20	18–65 years/all	Insulin resistance, quality-of-life assessment, liver function test, radiologic and serologic features of NASH, endoscopic ultrasound, and endoscopic sleeve gastroplasty	EUS-guided liver biopsy with portal pressure gradient measurement from 0–12 months	NASH, NAFLD, Fibrosis, and Obesity	NA/active, not recruiting	May 2021 to Dec. 2024	Trial contd.
NCT05486429	100	18–65 years/all	Abdominal ultrasound and lipid profile in different grades of NAFLD	Serum total cholesterol, triglyceride, HDL, LDL, and VLDL in 6 months	NAFLD	Observational/unknown	Jul. 2022 to Jan. 2023	Results submitted
NCT03307408	180	18 years and older/all	Blood collection	Assessment of liver stages by biochemical, genetic, and immunocytochemistry methods	HCC	Observational/unknown	Feb. 2017 to Feb. 2020	No results posted
NCT04264754	120	22 years and older/all	Blood collection	Methylation patterns-based blood test	HCC/cirrhosis	Observational/terminated	Feb. 2018 to Feb. 2021	Trial terminated
NCT05370053	450	18 years and older/all	ELF test	F3-F4 fibrosis over 2 years	NAFLD and fatty liver	NA/Unknown	Sep. 2020 to Dec. 2022	No results posted
NCT04785937	50	18 years and older/all	Ultrasound and magnetic resonance	False positives and negatives, sensitivity, and specificity	NAFLD, NASH, and liver fibrosis	NA/unknown	Jan. 2019 to Jun. 2022	No results posted
NCT05165446	35	18–99 years/all	Image-based surveillance	Tissue viscoelasticity assessment by novel MRE	NASH	Observational/active, not recruiting	Jan. 2022 to Mar. 2024	No results posted
NCT06328283	90	All/all	Glutamine synthetase and BCLAF1	Glutamine synthetase and BCLAF1 measurement	HCC	Observational/not yet recruiting	Apr. 2024 to Mar. 2026	Trial contd.

**Table 3 ijms-26-00306-t003:** Clinical trials with surgical procedure.

Clinical Trial #	# of Participants	Eligible Age/Sex	Intervention	Target	Conditions	Phase or Study Type/Status	Start and End Date	References
NCT03536650	14	28–75 years/All	Duodenal mucosal resurfacing	Adverse device effects over one year	NASH	NA/completed	Nov. 2017 to Dec. 2020	No results posted
NCT03524365	288	25–70 years/All	RYGB vs. SG with intensive lifestyle modifications	Histological resolution of NASH without worsening of fibrosis at 1 year after the interventions	NASH	NA/completed	Dec. 2018 to July 2022	[76]
NCT04677101	150	All/All	Liver biopsy	Validate blood biomarker sensitivity and accuracy in predicting NASH and fibrosis stage	NASH and Liver fibrosis	Observational/completed	Dec. 2020 to Apr. 2021	[76]
NCT05623150	710	18 years and older/All	Liver biopsy	Variations in metabolic gene markers	NAFLD, AFLD, NASH, and HCC/cirrhosis	Observational/not yet recruiting	Dec. 2022 to Mar. 2032	Trial contd.
NCT04281303	10	18–65 years/All	Endoscopic vertical gastroplasty	Adverse events, MELD, and Child–Pugh score	NASH/cirrhosis and obesity	Observational /unknown	Apr. 2020 to Apr. 2022	No results posted
NCT04653311	100	18–70 years old/All	Endoscopic sutured gastroplasty and endomina device	Rate of disappearance of NASH without worsening of fibrosis grade	NASH	NA/unknown	Jun. 2023 to Dec. 2023	No results posted

**Table 4 ijms-26-00306-t004:** Clinical trials with drug therapy.

Clinical Trial #	# of Participants	Eligible Age/Sex	Intervention	Targets	Conditions	Phase or Study Type/Status	Start and End Date	References
NCT05694923	55	18–70 years/all	Empagliflozin or diet control	Assessment of steatosis and fibrosis by FibroScan as a CAP score over 3 months	Non-diabetic NAFLD	NA/completed	Apr. 2023 to Apr. 2024	No results posted
NCT06256926	30	18–70 years old/all	Curcuvail or placebo	Change in NAFLD grading based on liver ultrasound from baseline to day 60	NAFLD	Phase 2/completed	Jan. 2021 to Nov. 2021	No results posted
NCT03439254	919	18 years and older/all	Obeticholic acid or placebo	Participants showing improvement in fibrosis by at least one stage without worsening of NASH	Compensated cirrhosis and NASH	Phase 3/completed	Aug. 2017 to Sep. 2022	Results posted
NCT05254626	100	18–65 years/all	Dapagliflozin and Pioglitazone	NAS score	NASH/diabetic/non-diabetic	Phase 2/completed	Aug. 2022 to Sep. 2024	No results posted
NCT04501406	166	21–75 years/all	Pioglitazone or placebo	Achieve an improvement of ≥2 points in NAS score without an increase in fibrosis stage.	NASH and Type 2 Diabetes Mellitus	Phase 2/recruiting	Dec. 2020 to Aug. 2027	Trial contd.
NCT04550481	45	18 years and older/all	Lisinopril MRI MRE LUE PDFF	PRO-C3, PC3X, steatosis, NFS, and inflammatory markers	HCC and NASH	Phase 2/ recruiting	May 2021 to Sep. 2025	Trial contd.
NCT05622071	50	18 years and older/all	Tislelizumab	ORR, limiting toxicity, OS, and PFS	HCC by BCLC stage	Phase 2/recruiting	Oct. 2023 to Apr. 2028	Trial contd.
NCT05391867	70	18–65 years/all	Lenvatinib vs. sorafenib	Overall Survival	HCC	NA/unknown	Jan. 2022 to Jun. 2023	No results posted
NCT05733897	150	All/all	Hydroxychloroquine	ALT levels over 1 year	NASH	Observational/recruiting	Jun. 2022 to Jul. 2025	Trial contd.
NCT06400771	12	19–55 years/male	DNP007	Safety, tolerability, pharmacokinetics, and immunogenicity of DNP007	NASH and Liver Transplant Rejection/Complications	Phase 1/recruiting	May 2024 to Dec. 2024	Trial contd.
NCT03479125	40	18 years and older/all	Ultrasound, MRI, and CT scan in HCC patients previously treated with Emricasan or placebo	Adjusted event rate for HCC in patients previously treated with Emricasan or placebo	Liver diseases	Observational/terminated because the production of Emricasan by the sponsor was discontinued	Feb. 2018 to Sep. 2019	Trial terminated
NCT03900429	1759	18 years or older/all	MGL-3196 (resmetirom) or placebo and liver biopsy	Two-point reduction in NAS score without worsening of fibrosis stage over 52 weeks	NASH	Phase 3/active, not recruiting	Mar. 2019 to Jan. 2028	[91]
NCT05500222	700	18 years or older/all	MGL-3196 (resmetirom) or placebo	Liver related outcomes with confirmed increase in MELD score <12 to ≥ 15	NASH/Cirrhosis	Phase3/recruiting	Aug. 2022 to Jan. 2027	Trial contd.

**Table 5 ijms-26-00306-t005:** Clinical trials with combination drug therapy.

Clinical Trial #	# of Participants	Eligible Age/Sex	Intervention	Targets	Conditions	Phase/Status	Start and End Date	References
NCT03449446	395	18 years to 80 years/all	Selonsertib, firsocostat, and cilofexor	Safety and tolerability (alone or in combination), change in liver fibrosis without worsening of NASH	Bridging fibrosis (F3) or compensated cirrhosis (F4)	Phase 2/completed	Mar. 2018 to Nov. 2019	[97]
NCT03517540	193	18 years or older/all	Tropifexor and cenicriviroc	Safety, tolerability, and efficacy of combination therapy, adverse and serious adverse event	NASH with fibrosis stage F2/F3	Phase2/completed	Sep. 2018 to Oct. 2020	[98]
NCT04065841	234	18 years or older/All	Tropifexor, licogliflozin, or placebo	At least 1-stage reduction in fibrosis without worsening of NASH	NASH with fibrosis	Phase2/terminated	Dec. 2019 to Oct. 2022	Terminated
NCT03987074	109	18 years to 75 years/All	Semaglutide, firsocostat, and cilofexor	Safety, tolerability of monotherapy and combination therapy	NASH with stage 2–3 fibrosis	Phase2/completed	July 2019 to July 2020	Completed
NCT05415722	162	18 years to 75 years/All	TERN-501 and TERN-101 or Placebo	Safety, efficacy, pharmacokinetics, and pharmacodynamics of mono or combination therapy	NASH/non-cirrhotic	Phase2/completed	July 2022 to July 2023	Completed
NCT02781584	220		Selonsertib, firsocostat, cilofexor, fenofibrate, and/or Vascepa	Safety and tolerability of drugs in patients with NASH/NAFLD	NASH/NAFLD	Phase2/completed	June 2016 to Dec. 2020	[99]

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
