# Peer review of "NAFLD and NAFLD Related HCC: Emerging Treatments and Clinical Trials"

_ijms, 2025, doi:10.3390/ijms26010306_

Round 1
Reviewer 1 Report (Previous Reviewer 1)
Comments and Suggestions for Authors
nothing to add
Author Response
We thank the reviewer for going through the revised manuscript and accepting the current version.
Reviewer 2 Report (New Reviewer)
Comments and Suggestions for Authors
Review paper entitled: „NAFLD and NAFLD Related HCC: Emerging Treatments and Clinical Trials“, describes new clinical studies regarding NAFLD. In general, I have no major objections. I only think that it is unnecessarily overextended, based on studies that are being carried out or have been completed but there are still no results. If the editors think that the number of pages is not a problem, the work can be accepted for publication.
Author Response
We thank the reviewer for thoroughly reviewing the latest revised version. Manuscript was revised based on previous comments and suggestions, which has added few extra pages. However, new version now includes critical analysis of the results from completed trials and addressing the issue of heterogeneity. Also, as suggested by the reviewer few additional references were added.
This manuscript is a resubmission of an earlier submission. The following is a list of the peer review reports and author responses from that submission.
Round 1
Reviewer 1 Report
Comments and Suggestions for Authors
Although the authors gave a useful summary of current clinical trials in MASLD, MASH, and HCC, their work could be improved with a more critical examination and wider contextualization of the results.
1. While the manuscript provided a comprehensive overview of ongoing trials, there is limited critical analysis of the results from completed trials. More in-depth discussion of the implications of these results would strengthen the review.
2. The review focused primarily on clinical outcomes but could benefit from more discussion of the underlying mechanisms of action for the various interventions.
3. The clinical trials' statistical techniques and the quality of the evidence offered by various study designs were not extensively discussed. Providing this information would aid readers in assessing the caliber of the supporting data.
4. To help identify the most promising directions for future research, the review could benefit from more direct comparisons between various interventions or approaches.
5. The manuscript's impact would be increased by a more thorough discussion of emerging trends or promising areas for further research, even though the conclusion mentioned some potential directions.
6. There is limited discussion of patient stratification or personalized medicine approaches in the context of MASLD/MASH/HCC treatment. This is an important consideration given the heterogeneity of these conditions.
7. The review could benefit from more discussion of how these interventions and trials apply to different populations globally, given the varying prevalence and risk factors for MASLD/MASH/HCC in different regions.
8. One potentially crucial topic to cover in the manuscript was the ethical issues in clinical trials for these conditions.
9. Recently, the nomenclature of NAFLD and NASH was changed to the new terminology, MASLD and MASH. Reflect this throughout the entire manuscript.
10. The following articles (PMID: 33575290, PMID: 38623613, PMID: 36103899, PMID: 36537018) provide recent, relevant research covering topics closely related to the manuscript's focus on MASLD, MASH, and HCC. They offer insights into epidemiology, screening, diagnosis, and emerging treatment that could strengthen the manuscript's discussion and analysis.
Reviewer 2 Report
Comments and Suggestions for Authors
please find attached
